# Ferrocement, Carbon, and Polypropylene Fibers for Strengthening Masonry Shear Walls

**DOI:** 10.3390/ma16134597

**Published:** 2023-06-26

**Authors:** Enea Mustafaraj, Marco Corradi, Yavuz Yardim, Erion Luga, Muhammed Yasin Codur

**Affiliations:** 1College of Engineering and Technology, American University of the Middle East, Egaila 54200, Kuwait; enea.mustafaraj@aum.edu.kw (E.M.); erion.luga@aum.edu.kw (E.L.); muhammed.codur@aum.edu.kw (M.Y.C.); 2Department of Mechanical and Construction Engineering, Wynne Jones Building, Northumbria University, Newcastle upon Tyne NE1 8ST, UK; 3Department of Civil and Environmental Engineering, The University of Edinburgh, Edinburgh EH9 3FG, UK; yyardim@exseed.ed.ac.uk

**Keywords:** ferrocement jacketing, polypropylene fibers, mortar coating, diagonal compression, strengthening, rehabilitation

## Abstract

This paper describes an experimental investigation into the feasibility of using ferrocement jacketing, polypropylene fibers, and carbon fiber reinforced polymer sheets (CFRP) to enhance the shear resistance of unreinforced brick masonry. The study involved testing 12 wall panels in diagonal compression, three of which were strengthened using each of the above-mentioned techniques. The results showed that all three strengthening techniques led to a significant improvement in the shear resistance and deformation capacity of the unreinforced walls. Furthermore, the results showed that the strengthened walls exhibited a significant improvement in shear resistance and deformation capacity by a factor of 3.3–4.7 and 3.7–6.8, respectively. These findings suggest that ferrocement jacketing is a viable and highly effective method for strengthening masonry structures. Test results can assist in the decision-making process to identify the most suitable design and retrofitting solution, which could indicate that not only new materials, but also traditional methods and materials (ferrocement) could be interesting and effective, also considering their lower initial cost.

## 1. Introduction

The largest part of the existing building stock is made of masonry. This “material” has been used for centuries in construction, and it is well known that masonry walls can adequately resist the static vertical compressive loads produced by occupancy, self-weight, and imposed loads in buildings. However, horizontal loading [1] (in-plane and out-of-plane), mainly produced by the seismic actions, typically generates tensile stresses in masonry, with respect to which masonry is particularly weak [2,3,4]. This leads to cracking and failures in masonry members (Figure 1). Masonry shear walls are a fundamental structural part in buildings [5,6,7]. These are typically located between the openings. Both the roof structure and the floors, usually made of timber or steel beams in historic buildings, rest on the shear walls.

The use of advanced composite materials (known under the acronym FRP, fiber reinforced polymers) to strengthen and rehabilitate old masonry structural members are widely known and demonstrated [8,9,10]. Numerous applications on shear walls [10,11,12,13,14], columns [15,16], vaults [17,18], arches [19,20], and lintels [21,22] have shown the efficiency of FRPs in seismic reinforcement. FRPs are usually in the form of pultruded laminates, grids, or fabrics. However, externally bonded FRP fabrics (applied using the wet lay-up installation method) are more interesting because FRPs exhibit a high tensile strength-to-weight ratio, high adaptation to irregularities of masonry surface, and overall ease of installation: such reinforcement applications can be carried out more rapidly in comparison to other traditional techniques (grout injections, reinforced concrete coatings, etc.) and hence reduce the construction and building closure time [10,11].

Polypropylene fibers are rarely used in the rehabilitation of civil structures. One of the earliest applications of polypropylene fibers on masonry was proposed by Yardim and Lalaj [23]: the shear walls reinforced with polypropylene mortar plaster exhibited a significant improvement in in-plane lateral strength of up to 412% when compared to the control specimens. Umair et al. [24] studied the use of polypropylene bands to reinforce small brickwork wallets against in-plane and out-of-plane loading: the aim was to verify if a low-cost material such as polypropylene, having a large ductility and deformation capacity, and a low tensile strength, was able to produce an interesting increase in the mechanical properties of brickwork masonry. It was demonstrated that the reinforced polypropylene band did not significantly increase the lateral load capacity, but it did produce a significant increase in the residual post-peak strength of the masonry wall panels.

More test results are reported in [25,26,27,28,29]: however, research mainly focused on the use of polypropylene fibers as a micro-reinforcement of masonry joints or as a method of reinforcement of masonry mortars [30,31], where polypropylene fibers are typically added to reduce shrinkage during drying.

The use of a steel mesh-reinforced mortar coating is an effective and well-known method to reinforce shear walls [32,33]. Typically, a 150 × 150 mm or 100 × 100 mm steel mesh, made of 6 mm or 8 mm diameter ribbed rebars, is embedded into a cement concrete coating applied to one or both sides of the walls of a building [34,35]. Figure 2 shows this “traditional” application. The application of a steel mesh-reinforced coating produces a significant increase in both the shear strength and elastic modulus. Mortar coatings are typically 50–80 mm in thickness and need to be applied after accurately removing the preexisting mortar plaster from the walls. However, the high increase in stiffness (up to 20 times the original shear modulus of the unreinforced masonry) may sometimes be problematic as it drastically affects the structural response of the masonry buildings under horizontal loading, also reducing the masonry deformation capacity [36].

In order to conjugate the need to increase the lateral load capacity of the shear walls without significantly affecting the deformation capacity and the shear modulus, this experimental work focuses on the use of a steel welded wire mesh, commonly known as hutch mesh, embedded into a reduced-thickness cement mortar coating. The objective of this research is to study low-cost solutions (ferrocement and polypropylene fibers) rarely adopted for structural interventions in seismic-prone areas and compare them with a more “advanced”, more expensive, and less “environmentally friendly” (CFRP) method.

## 2. Materials and Experimental Method

The main aim of the testing campaign was to investigate various strengthening methods that can be used to improve the lateral load capacity of the masonry shear walls. The experimental campaign was designed in such a way as to compare the effectiveness of polypropylene fibers, ferrocement jacketing, and carbon fibers.

The experimental campaign consisted of a total of 12 wall specimens that were tested in diagonal compression (shear testing). Nominal wall dimensions were 1200 × 1200 × 250 mm (height (h) × width (w) × thickness (t)). Each retrofit has been applied to 3 wall panels (a total of 9 wall panels for the 3 retrofitting methods investigated). Three wall panels were left unreinforced (control specimens) to study the effectiveness of the retrofitting method by comparison.

### 2.1. Unreinforced (URM) Walls

Three walls (W1, W2, and W3) were left unreinforced as control specimens. The wall panels were constructed using bricks and ASTM type “O” mortar, with the intention of replicating the composition of mortars found in existing old buildings. Mortar was prepared with a volumetric mix ratio of cement:lime:sand, of 1:2:9, representing traditional brick constructions. The mortar’s compressive and tensile strengths were measured, and the following results were found: compressive strength of 2.37 MPa and tensile strength of 0.31 MPa [37]. The panels were assembled using solid clay bricks with nominal dimensions of 250 × 120 × 55 mm and a cement mortar. For the solid clay bricks, compressive and tensile strengths of 24.03 and 4.53 MPa were recorded, respectively [38]. The construction was carried out at the laboratory by experienced masons utilizing the English bond brick pattern (English bond has one course of stretcher only and a course of the header above it). Following assembly, the panels were left to cure for 28 days before any testing or reinforcement procedures were carried out. The testing continued until the point of failure, which was determined to have occurred following the appearance of a main diagonal crack, followed by a sudden drop in the ultimate load.

### 2.2. Strengthened Walls

Nine wall panels were reinforced: the letter designations FC, PP, and CFRP were used to identify panels reinforced with the ferrocement method, polypropylene, and carbon FRP, respectively. The objective of reinforcing old masonry structures is to enhance their ability to withstand combined tensile and compressive forces, often generated by seismic events. As previously mentioned, several methods may be employed to mitigate the risks posed by natural disasters, as well as to increase the lateral load-bearing capacity and overall structural performance, thereby extending the URM structure’s service life. The following sections outline the selected reinforcement techniques.

#### 2.2.1. Ferrocement Jacketing (FC)

This technique involves attaching a double-layered galvanized steel mesh to both sides of the wall, as shown in Figure 3. The mesh is a welded steel wire with a square aperture of 12 mm. The technical specifications for the mesh are provided in Table 1. To fix the mesh to the wall, mechanical anchors and 15–20 mm thick mortar coating were used. Three walls have been retrofitted using this method (W4-FC, W5-FC, and W6-FC). The steel mesh was fixed using anchors, namely, threaded bolts with a diameter of 8 mm (M8) and a length of 70 mm, along with steel washers. These were mounted on previously drilled holes, having 10 mm wall plugs on the bricks, spaced at 30 cm intervals. The spacing of the connections may be slightly modified, depending on the brick arrangements, to ensure that the connection was made on the bricks and not on the mortar joint. On average, there were 12 connections/m^2^ to ensure a proper connection of the mesh to the wall. Care was taken during the process of mounting the steel mesh on the wall faces to ensure proper layering and to provide a clearance of 5–10 mm between the mesh and the bricks to fill with the mortar. The mortar mix was prepared using a volumetric ratio of Portland cement:sand of 1:4, and a water/cement ratio of 0.4. The average compressive and tensile strengths of the mortar coating were 19.30 MPa and 2.72 MPa, respectively.

#### 2.2.2. Polypropylene (PP) Reinforced Mortar Coating

This strengthening method involved the use of 12 mm long polypropylene fibers embedded into a 25 mm thick fiber-reinforced mortar coating on both sides of the walls (Figure 4). The mortar mixture comprises a sand and cement ratio of 1:1, adding 1.5% PP fibers in volume and a water/cement ratio of 0.5. The average compressive and flexural strengths of this type of mix were 41 MPa and 6.2 MPa, respectively. The amount of PP fibers was established as appropriate to provide sufficient tensile strength, as it does not adversely affect the workability of the mortar mixture. The fibers enhance the mechanical properties of the mortar, as well as its toughness, although they did not significantly impact its compressive strength. The technical specifications of the fibers are presented in Table 2. To prepare the mixture, the fibers were dry mixed with the sand and cement, after which water was added to produce a plaster mixture. Three walls were retrofitted using this method (W7-PP, W8-PP, and W9-PP).

#### 2.2.3. CFRP Epoxy-Bonded Sheet

The reinforcement pattern employed for the CFRP-strengthened specimens involved the application of a 300 mm wide unidirectional sheet onto a previously smoothed masonry surface (Figure 5). The CFRP sheets were applied along the wall diagonals on both sides. Glass fiber anchorages were additionally applied every 350 mm on previously drilled holes passing through the wall thickness. The installation of the CFRP sheet was carried out after drilling the anchorage holes and smoothing the wall surface. Initially, a layer of epoxy resin (commercial denomination: Sikadur 330) was applied to the wall surface. Subsequently, the CFRP sheet was installed, followed by another layer of epoxy resin, to ensure that the anchorages were firmly secured and to bind the fibrous reinforcement together. Technical specifications for both CFRP and epoxy can be found in Table 3.

### 2.3. Test Method

The wall panels were assembled within the laboratory. The ASTM E 519-07 [39] testing method was adopted to determine the diagonal tensile and shear strengths by compressing them along one wall diagonal until a diagonal tension failure occurred, causing the wall specimen to crack parallel to the direction of the diagonal load (Figure 6). The testing protocol stipulates that at least three like specimens should be tested, all constructed with the same size and type of masonry units, mortar, and workmanship. The tested specimens should be rotated by 45° and vertically loaded along one of the wall’s diagonals. However, due to the low masonry strength of the wall and the risk of inadvertently adding extra stress, the testing setup was modified. The wall specimen remained in its original position, and the loading mechanism was rotated accordingly. A movable test setup was created, consisting of two steel loading shoes placed on two diagonally opposite corners of the panel connected by four high-strength steel rods positioned along the compressed diagonal. A 50-tonne-capacity hydraulic jack was incorporated between the top loading shoe and a metallic plate connected to the steel rods, which developed tension forces on the four steel rods, compressing the wall diagonally, providing the desired failure mode, which was diagonal cracking and/or bed joint sliding failure. The applied load was gradually increased until failure occurred. Two diagonally positioned displacement gauges attached on each side wall panel over a gauge length of 1000 mm that were oriented parallel and perpendicular to the loading direction recorded the deformations of the wall specimen, including compression and elongation of diagonals. During the application of the diagonal compression test, the load distribution along the corners of the wall panels was carefully considered to avoid an excessive concentration of compressive stresses at the surface of metallic plates. The specimens were not moved for at least 7 days to achieve adequate curing and were stored in the laboratory for not less than 28 days.

The shear stress *S_s_* was estimated using:(1)$$S_{s}=\frac{0.707P}{A_{n}}$$
where *P*—load exerted along the compression diagonal; and *A_n_*—net sectional area of the wall specimen:(2)$$A_{n}=\frac{w+h}{2}t$$
where *w*—width of wall specimen (mm); *h*—height of specimen (mm); and *t*—total thickness of wall specimen (mm).

The shearing strain *γ* was calculated as:(3)$$\gamma =\frac{∆V+∆H}{g}$$
where ∆*V*—shortening (mm); ∆*H*—extension; and *g*—vertical gauge length (about 1000 mm) along the wall diagonal.

Finally, the shear modulus of rigidity *G* was estimated using:(4)$$G=\frac{S_{S}}{\gamma }$$

The shear modulus was calculated as the slope of a line connecting the origin and two specified points (70% and 33% of the wall shear strength) in the shear stress–shear strain diagram. These values have been identified with G_70%_ and G_33%_, respectively.

Finally, the drift ratio δ:(5)$$\delta =\frac{∆H}{h}$$

## 3. Numerical Modeling

The structural behavior of unreinforced and reinforced wall panels was numerically simulated (finite element (FE) analysis). Only ferrocement and polypropylene-strengthened specimens were considered in the numerical modeling. The FE analysis was conducted using the DIANA 9.6 [40] software package, which adopted a simplified micro-modeling approach for masonry modeling. The software was based on the displacement finite element method. The nonlinear analysis was carried out utilizing the cutback-based automated incremental procedure. During this process, the algorithm was designed to take as few load steps as possible to reduce the number of steps in the iterative procedure after defining the final loading. In the case of non-convergence, the load step was decreased, and the calculation was restarted. The Newton–Raphson method was utilized in this procedure.

To ensure a comprehensive comparison of the suggested strengthening methods, the numerical approach proposed by Zijl et al. [41] was employed. The modeling of the panels was achieved using a simplified modelling method with brick crack interface. This approach involved modeling bricks and mortar separately as two different materials using specific elements. For the bricks, the Q8MEM element was used, which is an isoperimetric, four-node plane stress element based on linear interpolation and Gauss integration, along with the CL12I interface element. The latter is an interface element between two lines in a two-dimensional configuration, specifically for the brick-joint and brick-crack interfaces (Figure 7) [40]. The mortar joint and the mortar/brick unit interface were lumped into a zero-thickness, discontinuous interface element that relates the normal stress (*σ_n_*) and shear stress (*τ*) to the normal interface displacement (*u*) and shear displacement (*v*), with a nominal width of 0.5 mm. In this model, eight plane stress elements and two interface elements were utilized. The mortar joints were the areas where cracks could be developed.

The brick units were modeled with continuum elements with elastic properties that were expanded to maintain the overall geometry of the masonry. In the middle of the brick, potential cracking interface elements were used. Furthermore, the non-linear behavior, produced by cracking, shear sliding, and crushing, was modeled using the interface elements.

One of the primary reasons for using the micro-modelling approach was its ability to replicate crack patterns and the complete load-displacement path of the masonry structure. This typically occurs in the mortar, which is weaker than the tile material of the bricks. It also provided a better understanding of the experimental results.

### 3.1. Adopted Crack-Shear-Crush (CSC) Interface Material Model

The different failure modes, such as joint tensile and sliding cracking, diagonal tensile cracking of the brick, and masonry crushing, are typically governed by three main criteria:Tension cut-off criterion;Coulomb friction criterion;Elliptical compressive cap criterion (Figure 8).

The interface elements’ friction and relative displacement vectors are represented by $\sigma ={\left[\begin{array}{cc} \sigma & \tau \end{array}\right]}^{T}$ and $\varepsilon ={\left[\begin{array}{cc} u & \nu \end{array}\right]}^{T}$, where σ and u are friction and relative displacement in the normal direction of the interface, whereas τ and ν are the friction and relative displacement in the perpendicular direction. The elastic stiffness matrix D, used in the elastic constitutive relationship σ=Dε is defined as:(6)$$D=\left[\begin{array}{cc} k_{n} & 0\\ 0 & k_{s} \end{array}\right]$$
where $k_{n}$ is stiffness in the normal direction, and $k_{s}$ is stiffness in the shear direction.

According to the tension cut-off criterion:(7)$$f_{1}=\sigma -\sigma_{t},$$
(8)$$\sigma_{t}=f_{t}e^{-\frac{f_{t}}{G_{f}^{I}}k_{1}}$$
where $\sigma_{t}$ is the tensile strength between brick and mortar, $f_{t}$ the bond strength, $G_{f}^{I}$ mode *I* fracture energy, and $k_{1}$ the equivalent plastic strain.

The Coulomb friction yield/crack initiation criterion is used to define shear-slipping. At this stage both adhesion softening and friction softening can be captured. The shear strength is proportional to the confining pressure with an initial offset (adhesion, *c*) and the angle of τ−σ with horizontal defines the friction angle, *Φ*. Cohesion will be zero, when the shear resistance decreases sufficiently, and the yield surface is defined as:(9)$$f_{2}=\left|\tau \right|+\sigma \Phi -c,$$
(10)$$c=c_{0}e^{-\frac{c_{0}}{G_{f}^{II}}k_{2}},$$
(11)$$\Phi =\Phi_{0}+\left(\Phi_{r}-\Phi_{0}\right)\frac{c_{0}-c}{c_{0}}$$
where Φ is the friction coefficient defined as the tangent of the friction angle Φ=tan(Φ), c is the adhesion, $\Phi_{0}$ and $\Phi_{r}$ are the initial and residual friction coefficients, respectively, $G_{f}^{II}$ is the mode II fracture energy, and $k_{2}$ is the equivalent plastic strain.

For the compressive cap criterion, the yield function for the compression cap is:(12)$$f_{3}=\sigma^{2}+C_{s}\tau^{2}-\sigma_{C}^{2}$$
where $C_{s}$ is a parameter that controls the shear stress contribution to failure and $\sigma_{C}^{2}$ the yield value.

The model, based on the micro-scale approach, was created in midas FX+ for DIANA 9.6. The mesh of the model was performed following three main stages: Firstly, the half-brick was created with interface elements to represent the brick crack and the brick joint, then the basic brick was duplicated in order to create the two-brick model with all the interface elements required for simulation (Figure 9). Lastly, the two-brick model was replicated in the horizontal and vertical direction to achieve the required wall dimensions of 1.2 × 1.2 m.

### 3.2. Assigning Material Properties

As the main aim of the modelling strategy was to investigate and compare the overall performance of the panels, some of the material parameters such as normal stiffness, *k_n_*, shear stiffness, *k_s_*, [42] and bond strength, *f_t_*, were found in the existing scientific literature; Van der Pluijm [43] conducted extensive research on the determination of mechanical behavior of brick–mortar interfaces, where bond strength, *f_t_*, mode I and mode II fracture energies were determined together with other parameters such as internal friction angle, dilatancy coefficient, etc.

The brick material and the brick-crack interface were kept linear, indicating that the cracks would be developed only in the mortar joints (as it was noted during the experimental stage of the campaign). The material properties are presented in Table 4. The bricks and mortar mechanical properties were determined experimentally as determined in Section 2. The three types of modeled panels have the same characteristics except for Young’s modulus, which was obtained from the experiments and is shown in Table 5.

### 3.3. Boundary Constraints

In order to effectively apply the in-plane load and to simulate the shear behavior of masonry, the bottom edges of the model were constrained in the horizontal and vertical directions, whereas for the top edges they were only for the vertical direction. Additionally, in order to prevent horizontal deformation of the upper edge, a multi-point constraint was applied.

The loading consisted of the application of a unit horizontal displacement at the top of the wall panel, which would be transferred uniformly along the entire upper edge due to the multi-point constraint previously applied.

The strengthened panels were modeled using an additional reinforcement layer made of a grid (in the case of ferrocement) and a plastering layer in the case of polypropylene fibers. The reinforcing material properties are summarized in Table 5.

## 4. Results and Discussion

The test results are given in terms of the mode of failure, crack pattern, mechanical parameters, shear stress–strain curve, and comparison of the numerical results with the experimental ones. These results provided a detailed panorama of the effectiveness of the proposed strengthening methods.

All tested specimens exhibited similar failure modes. Cracking was primarily recorded in the mortar bed and head joints, and failure was linked to the formation of a stair-like (zig-zag) crack along the diagonal of the wall specimen. The structural response of the panels could be classified as a failure due to diagonal tension. In the case of URM walls, failure was characterized by initial cracks followed by shear sliding along the cracked diagonal stepped joints (Figure 10).

Loading continued until the post-peak shear stress dropped abruptly from the maximum shear strength, and the resulting diagonal crack width became visible. The URM walls exhibited approximately linear behavior up to the first cracking, after which they suddenly failed along a diagonal step joint when they reached their diagonal tensile strength.

The experimental results demonstrate that strengthened surfaces in wall elements significantly contribute to load-carrying capacity and ductility in proportion to their connections with the wall surfaces. The reinforced surfaces effectively delay and confine cracking mechanisms that would occur due to shear-induced expansion in unreinforced walls. Moreover, by increasing the friction between wall elements, the reinforced surfaces help distribute the shear forces generated by loads more evenly, resulting in a substantial increase in both load and displacement capacities. These findings contribute to developing more effective and efficient strengthening techniques for wall elements.

In walls reinforced with ferrocement, the point where diagonal cracking begins was observed at roughly four times the maximum load of the URM control samples. This first significant diagonal cracking occurred at a point corresponding to approximately 70% of the peak load of the ferrocement-reinforced specimens. As the load approaches the peak, the number of diagonal cracks increases, and a ductile mode is observed by forming multiple cracks.

The resulting failure mode involved debonding between the mortar coating and the wall after the ultimate load. Due to high tensile stresses, connection failure occurred when the materials’ resisting capacities were exceeded, resulting in thick radial cracks around the unloaded upper and bottom edges of the panel. Despite the different final cracks observed in the panels, it was noted that the reinforcing coating had satisfactory behavior with respect to the strengthened panel. Furthermore, until the ultimate lateral-load capacity was reached, no debonding of the mesh and wall panel was detected (Figure 11). The hair-like cracks developed during the loading are marked in red color to enhance visualization.

In the case of polypropylene reinforcement, a similar situation was observed to some extent (Figure 12). However, because there was no mechanical connection between the two wall coatings, the composite structure dispersed earlier and lost its effectiveness more quickly than the ferrocement reinforcement. As a result of partial separation after reaching the ultimate load, a more brittle mechanism was observed.

In the test samples strengthened with CFRPs, the reinforcement was applied solely in a specific region to withstand the load applied in this experiment. A strengthening technique was employed that utilizes both mechanical connections with fiberglass FRP anchors and chemical connections with epoxy. This method has been shown to enhance the composite mechanism’s durability and efficiency. The CFRP used reached its ultimate capacity, and the failure mechanism began with the fracture of the CFRP.

The load-bearing capacity and mechanical behavior of reinforced walls were found to be governed by the mechanical properties of the CFRP used for reinforcement. Notably, unlike other test specimens strengthened with different methods, which exhibited strain-softening behavior, the CFRP-reinforced specimen did not display such behavior during testing.

The unique occurrence of a different crack mechanism in the ferrocement-strengthened walls may be attributed to the partial confinement of a pressure zone between the two steel shoes with the aid of anchors. This partial confinement is developed through additional friction at the head and bed joints of the wall bricks and shear anchor reinforcement connections, resulting in limited-width cracks in the confinement zone. Large cracks formed at the two ends of the remaining tensile zone immediately after the confined zone (Figure 13). This finding provides insight into the effectiveness of anchors in ferrocement-strengthened walls and highlights the importance of proper anchor location in the design of strengthened wall elements.

For the crack pattern of the reinforced panels, it was observed that the reinforcing coating had quite satisfactory structural behavior. Until the ultimate strength was reached, no debonding of the reinforcing mesh was observed. For such a composite structure, made of heterogeneous and anisotropic material, the most important properties are the ductility and the shear strength; thus, in such a case, the structural performance of this technique is deemed successful.

The experiment results (Table 6) indicate that the average shear strength of the URM panels was 0.133 MPa, with a maximum of 0.153 MPa reached for W3 and a minimum of 0.117 MPa for W2. The deformation capacity of these panels was limited to 0.336%, with a maximum of 0.384% for W1 and a minimum of 0.281% for W3. Figure 14 shows the shear stress vs. shear strain plot for all the tested wall specimens.

The panels strengthened with ferrocement jacketing exhibited an average shear strength of 0.728 MPa, which is 4.755 times higher than the control specimen. The maximum shear strength was observed in W6-FC, 0.822 MPa, and the minimum in W4-FC, 0.657 MPa. The deformation capacity of these panels was exceptional, having an average of 2.286%, with a maximum of 2.676% for W5-FC and a minimum of 1.872% for W4-FC. When compared to the control specimen, the improvement is 6.805 times.

Similarly, the panels strengthened with polypropylene-reinforced mortar reached an average shear strength of 0.509 MPa, with a maximum of 0.564 MPa for W8-PP and a minimum of 0.470 MPa for W7-PP. When compared to the control specimen, the improvement is 3.323 times. However, the deformation capacity was limited to 0.299%, which was smaller than the control specimen. As was observed from the failure pattern, the sudden diagonal crack caused the failure of the wall just after reaching the peaking load.

Furthermore, CFRP-reinforced panels showed a considerable improvement in both shear strength and deformation capacity, with an average of 0.591 MPa and 1.245%, respectively. W11-CFRP and W12-CFRP both reached a shear strength of 0.611 MPa, whereas W10-CFRP achieved a slightly lower value of 0.552 MPa. In terms of deformation capacity, W11-CFRP reached a value of 1.813%, whereas W10-CFRP had the lowest value of 0.889%. When compared to the control specimen, the improvement in the shear strength and deformation capacity were 3.862 and 3.706 times, respectively.

It is also interesting to study the development of shear modulus G (this has been calculated at 70% and 33% of the wall shear strength). It can be observed that the stiffness of all strengthened walls increased significantly when comparing them to a URM specimen (G_33%_ increased between 69 and 189% compared to URM walls). However, this enhancement in stiffness is progressively reduced by increasing the shear load: this can be noted by looking at G_70%_ values. These two indices (G_33%_ and G_70%_) can be intended to represent the progression of the damage in the masonry material and the slippage at the bond between masonry and retrofit. In an ideal material, where both masonry and retrofit remain in the elastic phase, and their bond is perfect, the values of G should remain unchanged up to failure. The fact that the secant stiffness G decreases when calculated for higher values of the shear loads clearly indicates that phenomena of local cracking occur in the masonry (most likely in the mortar bed joints), and debonding develops in these regions. In this situation, the strengthened walls were still able to resist the diagonal loading by redistributing the tensile stresses from masonry to the retrofit, as common in statically indeterminate structures (known as progressive collapse).

### Numerical vs. Experimental Results

Comparing the experimental results with the numerical ones is a challenging and problematic task: clearly, numerical results are governed by the mechanical parameters given in Table 4 and Table 5. These values have been found in the scientific literature, with no direct relation to the mechanical properties of the brickwork masonry used in this experimental work. This is a significant limitation of this investigation. However, the aim of the numerical analysis was to verify if the use of a simplified numerical approach, only based on mechanical parameters found in the literature, was able to provide acceptable results useful in structural design and present the professional world with an indication of the error this very simplified numerical method can produce.

Figure 15 and Figure 16 show the comparison between numerical and experimental results in terms of shear behavior. It can be noted that a large shear strength error was noted for unreinforced and PP-reinforced walls (about 30%), while for FC-reinforced walls, the error was significantly smaller (about 15%).

The ferrocement-strengthened specimens exhibited a higher shear stress of 0.70 MPa and a total shear strain of about 0.025. The stress–strain diagram obtained after nonlinear analysis showed that ferrocement-strengthened specimens achieved the highest shear stress of load of 0.595 MPa and a maximum strain of 0.0159, considerably higher than the other two panels (Figure 15).

Polypropylene-strengthened panels exhibited similar behavior in both cases; high shear stress (0.480 MPa) but very low shear strain value (0.0015), and in the numerical analysis, a maximum shear stress of 0.332 MPa and a maximum strain of 0.00121. URM panels exhibited a lower shear stress of 0.233 MPa and a maximum strain of 0.0011.

Figure 16 shows the comparison between the wall specimens. It was observed that for all specimens, the numerical analysis provided lower shear stresses and strains compared to the experimental results emphasizing the fact that some of the assumed parameters were more conservative.

Due to the irregular nature of masonry, its non-isotropic properties, and the fact that it is assembled using non-industrial techniques, the simplified numerical method employed in this study may appear to yield reasonable quantitative results. However, it is important to note that these results may not hold up to rigorous qualitative standards. Conducting a more precise numerical analysis, such as employing micro-modelling techniques that separately model all the materials involved (including mortars, blocks, and retrofits), could potentially provide greater accuracy. Nevertheless, it may be impractical to employ such detailed modeling approaches in the context of professional practice, and design and retrofit interventions.

## 5. Conclusions

In this paper, the results of a series of experimental tests on brick masonry walls strengthened with different methods and materials are presented. This study investigates the effectiveness of new retrofitting techniques on shear wall panels. Twelve solid brickwork masonry wall panels (1200 × 1200 × 250 mm) were tested for in-plane static loads.

The results of this investigation suggest the following conclusions:The behavior of the unreinforced shear walls was highly influenced by the strength of the mortar used in construction. In fact, only one type of failure was observed, namely, mortar cracking and debonding of the mortar from the bricks during shear testing of URM walls. Shear walls displayed considerable post-elastic deformation and energy dissipation and behaved in a quasi-ductile manner.Three different types of shear reinforcement were used and tested: (1) externally epoxy bonded CFRP sheets, (2) short polypropylene fibers embedded into a mortar coating, and (3) mortar jacketing reinforced with steel-wire mesh (ferrocement);Strengthening of unreinforced shear walls by the three methods contributed significantly to the shear performance of the walls, both increasing the lateral-load performance, shear stiffness, and ductility; the application of the different retrofits did not drastically change the wall’s failure mode: mortar in the head and bed joints cracked during shear testing, but the application of the surface retrofits could significantly produce a bridging effect to delay crack propagation in masonry. This has led to a substantial improvement of the lateral-load capacity and an ability to withstand the lateral load for higher levels of the wall’s shear deformation;According to the experimental findings, the ferrocement-strengthened panels exhibited a notable 546% increase in shear strength and a remarkable 680% improvement in deformation capacity compared to the control specimens. In contrast, the polypropylene-reinforced panels demonstrated a 382% enhancement in shear strength; however, they could only achieve 80% of the deformation capacity of the control specimens. The CFRP-reinforced panels exhibited a significant 444% increase in strength and a notable 370% improvement in ductility when compared to the unreinforced panels.To further investigate the performance of these techniques, a simplified numerical modeling was performed using commercially-available DIANA FEA software. It was noted that the numerical procedure was able to capture the structural response of both unreinforced and reinforced wall panels with acceptable reliability. The finite element analysis produced conservative results, with ferrocement exhibiting a 300% improvement in strength and an impressive 722% increase in ductility. In contrast, polypropylene showed a 200% enhancement.

It is suggested that incorporating different phases for masonry material and the attached retrofits, as well as the bond connection between them in an incremental finite element analysis, may lead to a closer correlation with experimental test results. Numerical research is currently underway to investigate these matters.

Readers should be aware of the drawbacks of the proposed retrofitting methods. The main limitation on the fullest possible use of small-diameter steel wires in ferrocement retrofit is the risk of corrosion. However, this could be mitigated by using high-strength stainless steel grids. For CFRP and PP applications, fiber and matrix degradation in exposed environments (UV radiation, extreme and harmful weather events, rain, and humidity) could be significant. More tests will be necessary to assess the long-term effectiveness of the proposed retrofitting methods.

## Figures and Tables

**Figure 1 materials-16-04597-f001:**
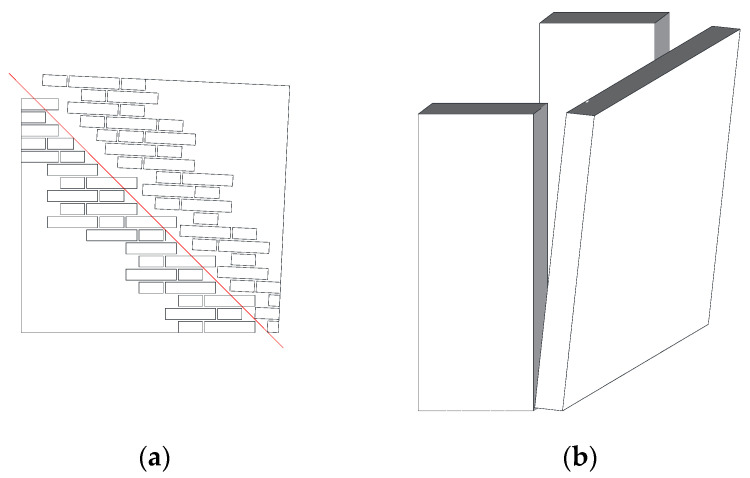
Common failure modes of masonry members under horizontal loading: (**a**) in-plane failure; (**b**) out-of-plane rocking mechanism.

**Figure 2 materials-16-04597-f002:**
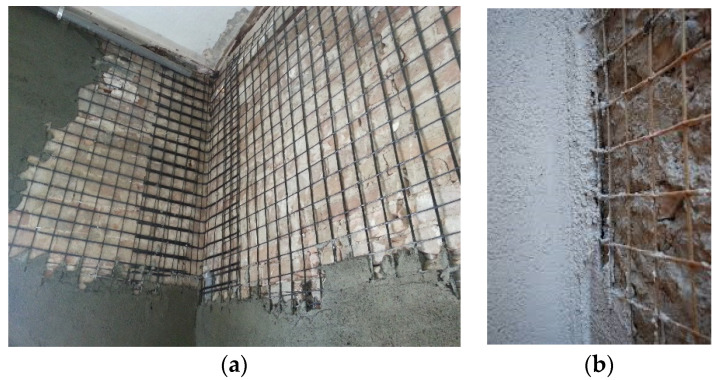
Steel mesh reinforcement of shear walls: the mesh is embedded into a cement mortar coating applied to one or both sides of the shear walls. The mortar coatings are typically connected to each other with transversal ribbed steel rebars inserted into holes drilled into the wall: (**a**) Reinforcement photo; (**b**) detail.

**Figure 3 materials-16-04597-f003:**
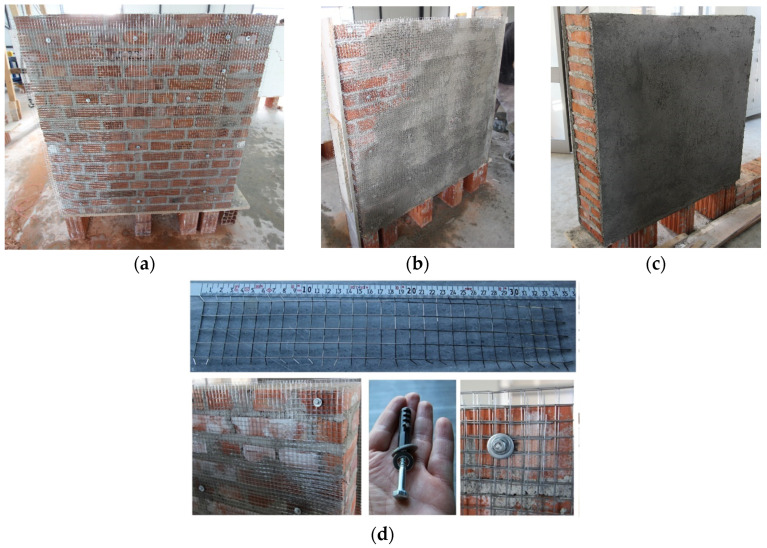
The plastering with ferrocement jacketing: (**a**) installment of the mesh; (**b**) application of the first plaster layer; (**c**) application of the second layer; (**d**) details of the mesh, connector, and the mesh-wall connection.

**Figure 4 materials-16-04597-f004:**
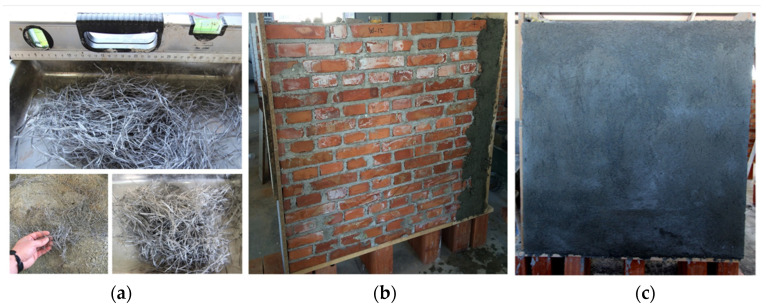
The plastering process with polypropylene fibers: (**a**) polypropylene fibers; (**b**,**c**) before and after application to brickwork walls.

**Figure 5 materials-16-04597-f005:**
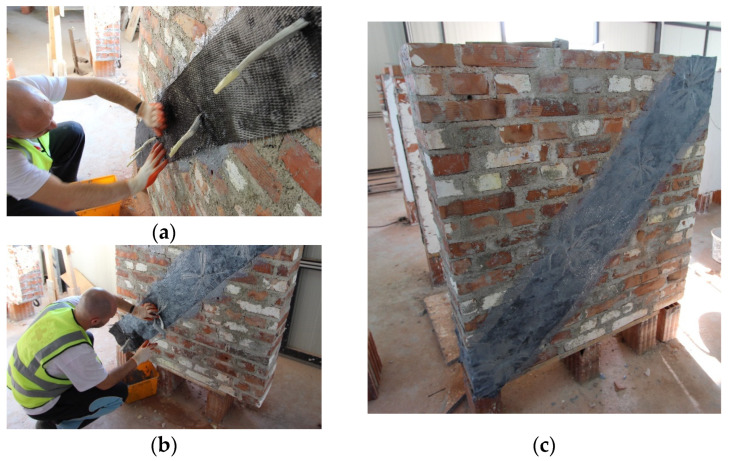
Application of carbon fiber reinforced polymer (CFRP): (**a**) placing of the CFRP sheet; (**b**) application of epoxy; (**c**) finished panel.

**Figure 6 materials-16-04597-f006:**
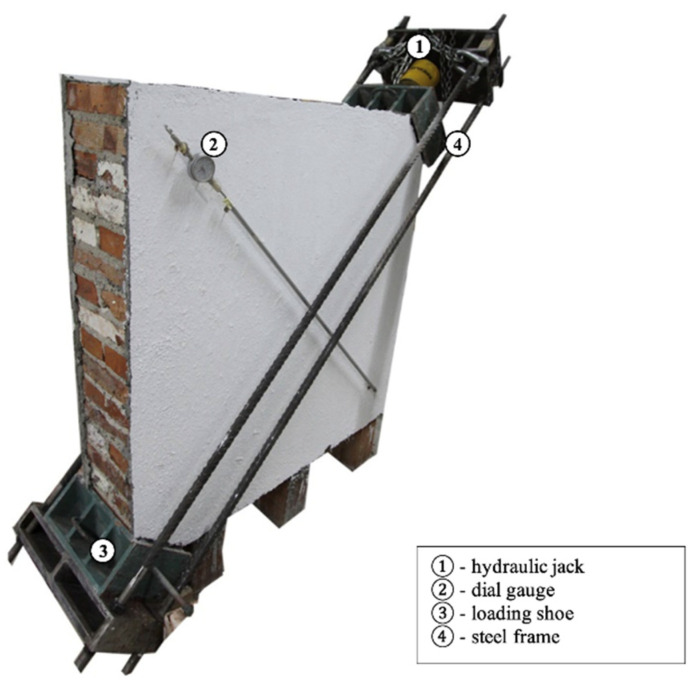
Diagonal compression test set-up.

**Figure 7 materials-16-04597-f007:**
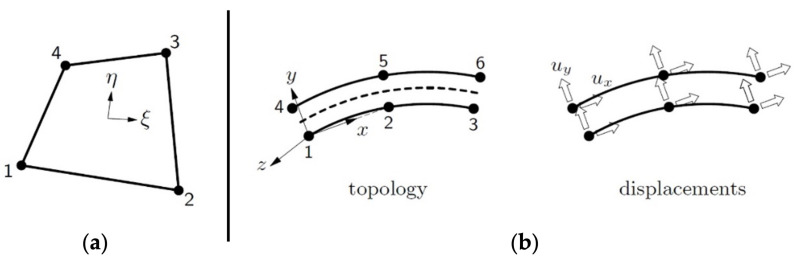
DIANA 9.6 elements used for modeling: (**a**) Q8MEM, plane stress element; and (**b**) CL12I, interface element [40].

**Figure 8 materials-16-04597-f008:**
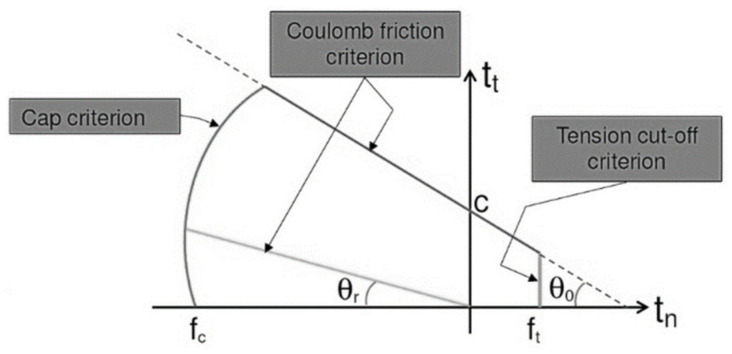
Interface model used in this study.

**Figure 9 materials-16-04597-f009:**
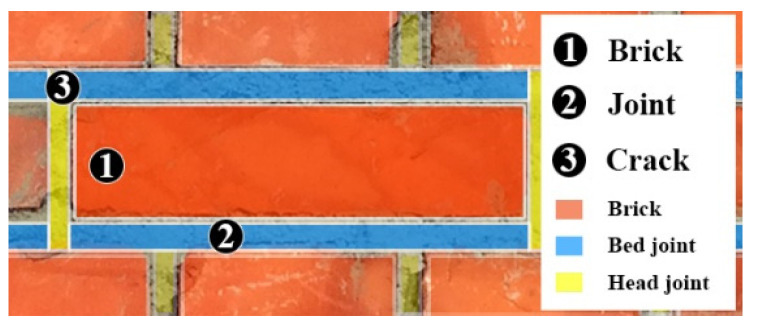
Partition of the masonry into elements ready for modeling.

**Figure 10 materials-16-04597-f010:**
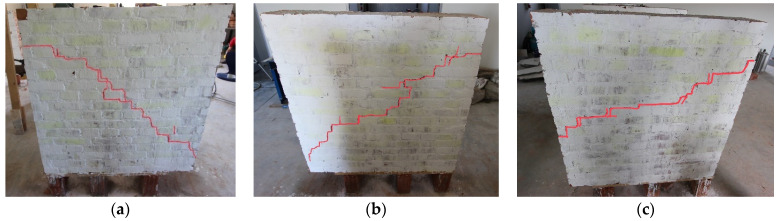
The failure mode for URM panels: (**a**) W1; (**b**) W2; (**c**) W3.

**Figure 11 materials-16-04597-f011:**
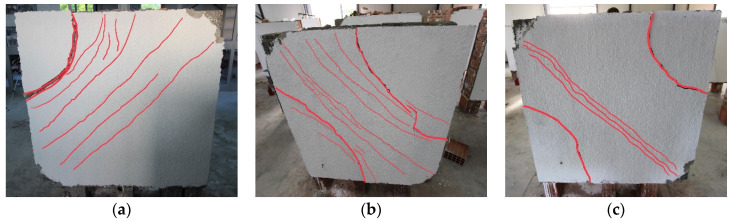
The failure mode of FC-strengthened panels: (**a**) W4-FC; (**b**) W5-FC; (**c**) W6-FC.

**Figure 12 materials-16-04597-f012:**
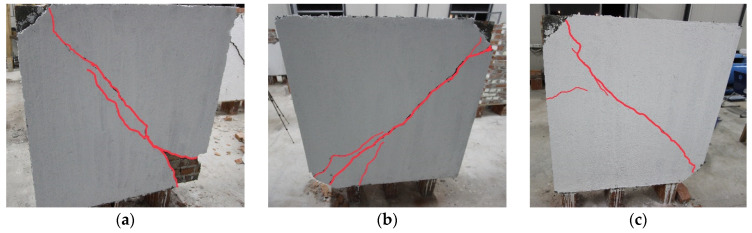
The failure mode of polypropylene strengthened wall panels: (**a**) W7-PP; (**b**) W8-PP; (**c**) W9-PP.

**Figure 13 materials-16-04597-f013:**
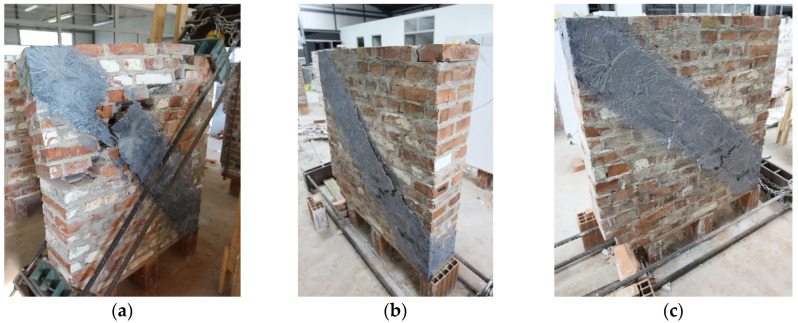
The failure mode of wall panels reinforced with carbon fibers: (**a**) W10-CFRP; (**b**) W11-CFRP; (**c**) W12-CFRP.

**Figure 14 materials-16-04597-f014:**
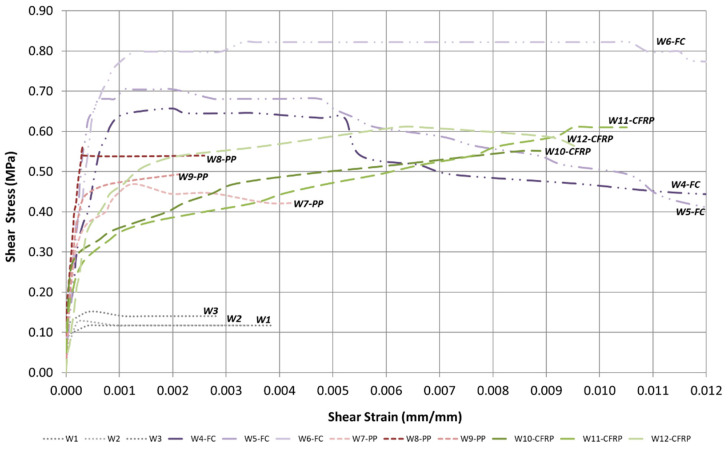
Stress–strain response of all the tested panels.

**Figure 15 materials-16-04597-f015:**
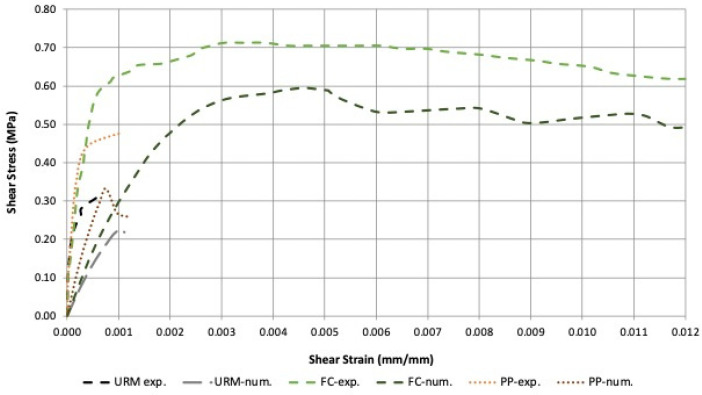
Summary of stress–strain response of experimental and numerical results.

**Figure 16 materials-16-04597-f016:**
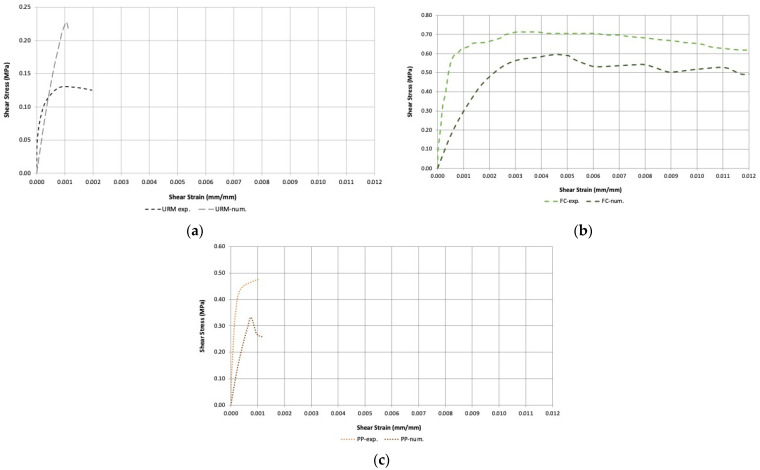
Comparison between experimental and numerical stress–strain curves: (**a**) unreinforced walls; (**b**) carbon fiber reinforcement; (**c**) polypropylene fiber reinforcement.

**Table 1 materials-16-04597-t001:** Technical specifications for steel wire mesh (from producer data sheet).

Mesh Type	Galvanized Welded Wires
Mesh size (mm)	12 × 12
Nominal wire diameter (mm)	1
Weight (kg/m^2^)	0.3
Young’s modulus (GPa)	170
Yield strength (MPa)	200
Ultimate strength (MPa)	550

**Table 2 materials-16-04597-t002:** Technical specifications of polypropylene fibers (from manufacturer).

Chemical Base	100% Polypropylene Fiber
Specific gravity (g/cm³)	0.91
Fiber length (mm)	12
Fiber diameter (mm)	18
Melting point (°C)	160
Fiber tensile strength (MPa)	300–400
Fiber Young’s modulus (MPa)	~4000
Specific surface area of fiber (m^2^/kg)	250

**Table 3 materials-16-04597-t003:** Mechanical properties of CFRP sheet (from producer data sheet, SikaWrap-230 C).

Fiber Type	Carbon
Orientation	unidirectional
Fiber dry weight density (g/m^2^)	230
Fiber tensile strength (MPa)	4300 *
Fiber Young’s modulus (GPa)	238 *
Fiber elongation at break (%)	1.8
Epoxy resin tensile strength (MPa)	30
Epoxy resin flexural elastic modulus (GPa)	3.8
Epoxy resin tensile elastic modulus (GPa)	4.5

* These mechanical values were calculated using an “equivalent thickness” of the unidirectional carbon sheet (0.129 mm).

**Table 4 materials-16-04597-t004:** Material properties used for the model [40].

Masonry	Young’s Modulus E (MPa)	Shown in Table 5
	Poisson’s ratio n (−)	0.15
	Linear normal stiffness D_11_ (N/mm^3^)	10^4^
Cracks	Linear tangential stiffness D_12_ (N/mm^3^)	10^3^
	Linear normal stiffness D_11_ (N/mm^3^)	83.0
	Linear tangential stiffness D_12_ (N/mm^3^)	36.0
	Mortar tensile strength f_t_ (MPa)	0.268
	Fracture energy G_f_ (N/mm)	0.018
	Cohesion c (MPa)	0.35
	Friction angle tan φ	0.75
	Dilatancy angle tan ψ	0.60
Joints	Residual friction coefficient Φ	0.75
	Confining normal stress for ψ_0_, σ_u_ (MPa)	−1.3
	Exponential degradation coefficient δ	5.0
	Mortar compressive strength f_c_ (MPa)	2.816
	Shear traction control factor C_s_	9.0
	Compressive fracture energy G_fc_ (N/mm)	5.0
	Equivalent plastic relative displacement K_p_	0.093
	Fracture energy factor b	0.05

**Table 5 materials-16-04597-t005:** Mechanical parameters used for unreinforced and reinforced wall panels.

	Masonry Young’s Modulus E (MPa)	Masonry Shear Modulus G (MPa)
URM	530	212
Ferrocement	1218	487
Polypropylene	1265	506

**Table 6 materials-16-04597-t006:** Test results.

Wall Panel	P_max_ (kN)	S_s_ (MPa)	δ (%)	G_70%_ (MPa)	G_33%_ (MPa)
W1	49.8	0.117	0.384	68	415
W2	54.8	0.129	0.343	264	1096
W3	64.8	0.153	0.281	305	1620
W-average	56.5	0.133	0.336	212	1044
W4-FC	279.0	0.657	1.872	567	1508
W5-FC	299.0	0.704	2.676	646	1759
W6-FC	348.7	0.822	2.311	248	2025
FC-average	308.9	0.728	2.286	487	1764
*FC* vs. *URM*		5.469	6.805	2.29	1.69
W7-PP	199.3	0.470	0.426	368	2657
W8-PP	239.1	0.564	0.259	916	3985
W9-PP	209.2	0.493	0.211	234	2405
PP-average	215.9	0.509	0.299	506	3016
*PP* vs. *URM*		3.822	0.888	2.38	2.89
W10-CFRP	234.2	0.552	0.889	65	3346
W11-CFRP	259.1	0.611	1.813	34	960
W12-CFRP	259.1	0.611	1.034	59	1036
CFRP-average	250.8	0.591	1.245	53	1781
*CFRP* vs. *URM*		4.443	3.706	0.25	1.71

## Data Availability

Data are available on request due to restrictions, e.g., privacy or ethical. The data presented in this study are available on request from E.M.

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
