# Peer review of "Ferrocement, Carbon, and Polypropylene Fibers for Strengthening Masonry Shear Walls"

_materials, 2023, doi:10.3390/ma16134597_

Round 1

Reviewer 1 Report

1.In Figure 2-a, it can be seen that the threaded bolts are not evenly arranged, does it affect the test results?

2. Part of the picture number is wrong, as shown in Figure 90, 101, 125 and 136;

3.From Section 4 of the article, it can be seen that the test has been simulated in advance, but why there is no simulation effect diagram, especially the corresponding cracking simulation diagram. Is the conclusion given too simple? The simulation effect is not ideal, so what is the meaning of this part?

4.In Figure 14, the URM analysis is "the average shear strength of the URM panels was 0.153 MPa, with a maximum of 0.177 MPa reached for W1 and a minimum of 0.129 MPa for W2.", but the maximum shear stress of the URM specimen in Figure 14 shows W3, and the maximum test load in Table 6 is also W3, which is inconsistent with the description, and needs to be verified

5.It can be seen from Figure 14 that the results of the same group of specimens show a certain degree of discreteness, but the results of different groups of specimens have certain regularity. Among the three specimens in the same group, the results of two specimens show consistency, but there is always one showing a large deviation from the others. Are there other factors in the test?

6. In Figure 14, W-8 value reaches its peak and remains stable. Is it wrong?

7.The analysis of the reinforcement mechanism is still lacking, and the feasibility of reinforcement is only given from the comparison of test phenomena and data levels, and the depth of content is insufficient.

The English writing level of the article is relatively high, which can well display the viewpoints of the paper, and the layout is relatively reasonable.

Author Response

1.In Figure 2-a, it can be seen that the threaded bolts are not evenly arranged, does it affect the test results?

RESPONSE: The arrangement of the bolts is made in a certain way that they are anchored in the bricks. Overall, the bolts will ensure the connection of the mesh in the total area of the wall panel. As long as there is a sufficient number of them (12 anchors per square meter) all over the reinforced area, it will not have any effect on the performance.

  1. Part of the picture number is wrong, as shown in Figure 90, 101, 125 and 136;

RESPONSE: Corrected.

3.From Section 4 of the article, it can be seen that the test has been simulated in advance, but why there is no simulation effect diagram, especially the corresponding cracking simulation diagram. Is the conclusion given too simple? The simulation effect is not ideal, so what is the meaning of this part?

RESPONSE: As we know, masonry is a non-homogeneous anisotropic material whose real behavior is very difficult to be predicted. The FE simulation aimed at analyzing the behavior, and as it was observed from the results, the shear-stress strain behavior was very similar in terms of deformation capacity and the shear strength,

4.In Figure 14, the URM analysis is "the average shear strength of the URM panels was 0.153 MPa, with a maximum of 0.177 MPa reached for W1 and a minimum of 0.129 MPa for W2.", but the maximum shear stress of the URM specimen in Figure 14 shows W3, and the maximum test load in Table 6 is also W3, which is inconsistent with the description, and needs to be verified.

RESPONSE: Thank you for pointing this out. It was a typo. The value for W1 is corrected to 0.117 instead of 0.177 MPa. At the same time, also the ratios of FC vs. URM, PP vs. URM and CFRP vs. URM are also updated.

5.It can be seen from Figure 14 that the results of the same group of specimens show a certain degree of discreteness, but the results of different groups of specimens have certain regularity. Among the three specimens in the same group, the results of two specimens show consistency, but there is always one showing a large deviation from the others. Are there other factors in the test?

RESPONSE: During the experimental campaign, we tried to keep all the input parameters (construction materials, mortar mix, using the same mason worker) as well as the testing conditions the same. The variation in the results is due to the non-homogeneity and anisotropy of masonry. Nevertheless, the maximum variance is seen in URM (c.o.v= 0.13), then in FC (c.o.v = 0.12), PP (c.o.v = 0.1) and CFRP (c.o.v = 0.06)

  1. In Figure 14, W-8 value reaches its peak and remains stable. Is it wrong?

RESPONSE: W8 was reinforced with Polyproplyene. The panel managed to keep the ultimate load constant for a certain time, then it failed. Considering its deformation capacity, slightly less than 0.003 mm/mm deformed more than W9 but less than W7.

  1. The analysis of the reinforcement mechanism is still lacking, and the feasibility of reinforcement is only given from the comparison of test phenomena and data levels, and the depth of content is insufficient.

RESPONSE: Thank you for this comment. Analysis was elaborated more and reflected in the text

Reviewer 2 Report

The paper is well-written and shows promise for publication. However, there are several major aspects that need to be addressed before it can be considered for publication.

Firstly, the introduction needs to clearly identify the gap in knowledge that this paper fills. It should be organized as follows: provide a general overview of the problem, discuss what has already been done in the field, highlight the aspects that are lacking in the existing literature, and clearly explain how the authors' work addresses these gaps.

Additionally, the reference list is missing several important contributions related to the experimental side of masonry walls, both with and without reinforcement. The following references should be included:

Khan, N. A., Aloisio, A., Monti, G., Nuti, C., & Briseghella, B. (2023). Experimental characterization and empirical strength prediction of Pakistani brick masonry walls. Journal of Building Engineering, 71, 106451.

Gonen, S., Pulatsu, B., Lourenço, P. B., Lemos, J. V., Tuncay, K., & Erduran, E. (2023). Analysis and prediction of masonry wallette strength under combined compression-bending via stochastic computational modeling. Engineering Structures, 278, 115492.

There is an issue with the formatting in line 155, which should be corrected.

To provide clarity on the tests conducted, it would be beneficial to create a table that lists all the tests, their labels, and the variables that have been varied. This table should also distinguish between different specimens used in the study.

The comparison between the experimental and numerical results could be improved by creating a table that reports the numerical values alongside the corresponding experimental values for initial stiffness, secondary stiffness, capacity, and ductility. Additionally, the table should include the relative error between the two sets of values.

Lastly, the language and spelling in the paper need improvement as there are several typos present. Careful proofreading and editing should be conducted to enhance the overall quality of the manuscript.

Author Response

Firstly, the authors would like to thank the reviewers for their valuable comments and suggestions. We believe that thanks to these suggestions, the quality of the manuscript is considerably improved.

Reviewer #2

The paper is well-written and shows promise for publication. However, there are several major aspects that need to be addressed before it can be considered for publication.

  • Firstly, the introduction needs to clearly identify the gap in knowledge that this paper fills. It should be organized as follows: provide a general overview of the problem, discuss what has already been done in the field, highlight the aspects that are lacking in the existing literature, and clearly explain how the authors' work addresses these gaps.
  • Additionally, the reference list is missing several important contributions related to the experimental side of masonry walls, both with and without reinforcement. The following references should be included:

Khan, N. A., Aloisio, A., Monti, G., Nuti, C., & Briseghella, B. (2023). Experimental characterization and empirical strength prediction of Pakistani brick masonry walls. Journal of Building Engineering, 71, 106451.

Gonen, S., Pulatsu, B., Lourenço, P. B., Lemos, J. V., Tuncay, K., & Erduran, E. (2023). Analysis and prediction of masonry wallette strength under combined compression-bending via stochastic computational modeling. Engineering Structures, 278, 115492.

RESPONSE: these relevant citations have been added to the manuscript and the objective of this study is added in the Introduction.

  • There is an issue with the formatting in line 155, which should be corrected.

RESPONSE: Corrected.

  • To provide clarity on the tests conducted, it would be beneficial to create a table that lists all the tests, their labels, and the variables that have been varied. This table should also distinguish between different specimens used in the study.

RESPONSE: The text has been improved to better illustrate the test program “The main aim of the testing campaign was to investigate various strengthening methods that can be used to improve the lateral load capacity of the masonry shear walls. The experimental campaign was designed in such a way as to compare the effectiveness of polypropylene fibers, ferrocement jacketing, and carbon fibers used as strengthening techniques. The experimental campaign consisted of a total of 12 wall specimens that were tested in diagonal compression. Nominal wall dimensions were 1200x1200x250 mm (height (h) x width (w) x thickness (t)). Each retrofit has been applied to 3 wall panels (total 9 wall panels for the 3 retrofitting methods investigated). 3 wall panels were left unreinforced (control specimens) to study the effectiveness of the retrofitting method by comparison.”

  • The comparison between the experimental and numerical results could be improved by creating a table that reports the numerical values alongside the corresponding experimental values for initial stiffness, secondary stiffness, capacity, and ductility. Additionally, the table should include the relative error between the two sets of values.

RESPONSE: we have completely revised by adding text and analysis to the section where numerical results are compared with experimental ones.

Lastly, the language and spelling in the paper need improvement as there are several typos present. Careful proofreading and editing should be conducted to enhance the overall quality of the manuscript.

RESPONSE: Corrected. Thank you

Reviewer 3 Report

The authors present an interesting research work that may be useful for technicians and professionals in the construction sector. Full-scale walls are tested, which gives a very approximate idea of the mechanical behaviour of these construction systems with and without reinforcement. The article is appropriate for this journal.

However, some comments need to be resolved before publication:

Line 15: "This paper..." suggested to be changed to "This research..."

Line 34, add reference to support that statement.

Line 72, 150 x 150 mm

The objective is unclear and diffuse. Clearly state the objective of the research at the end of the introduction.

Page 5 has a page break

Line 155 correct, there appears to be a figure that has been moved.

Figure 4 is repeated

Section 3 and 4 should be merged with section 2.Materials and Methods.

Section 4.1. is well known, perhaps it could be schematised.

Line 336 and 338, cannot be Table 1, should be Table 5.

Line 379, Correct Figure 101) and the end of the sentence.

There is no final discussion of the numerical results and at no point are the results compared with other research. This section should be corrected.

Line 490. Figure 136

The limitations found in this research should be included at the end of the conclusions.

-The article is generally well written.

Author Response

Firstly, the authors would like to thank the reviewers for their valuable comments and suggestions. We believe that thanks to these suggestions, the quality of the manuscript is considerably improved.

Reviewer #3

The authors present an interesting research work that may be useful for technicians and professionals in the construction sector. Full-scale walls are tested, which gives a very approximate idea of the mechanical behaviour of these construction systems with and without reinforcement. The article is appropriate for this journal.

However, some comments need to be resolved before publication:

  • Line 15: "This paper..." suggested to be changed to "This research..."

RESPONSE: Corrected. Thank you

  • Line 34, add reference to support that statement.

RESPONSE: Corrected. A reference has been added here to support this.

  • Line 72, 150 x 150 mm

RESPONSE: Corrected. Thank you

  • The objective is unclear and diffuse. Clearly state the objective of the research at the end of the introduction.

RESPONSE:  Thank you for this question. The following paragraph was added: “The objective of this research is to study low-cost solutions (ferrocement and polypropylene fibers) rarely adopted for structural interventions in seismic prone areas and compare them with a more “advanced” method, more expensive and less “environ-mentally friendly” (CFRP).” 

  • Page 5 has a page break

RESPONSE: Corrected. It was a formatting issue. Thank you

  • Line 155 correct, there appears to be a figure that has been moved.

RESPONSE:  Corrected. It was a formatting issue. Thank you

  • Figure 4 is repeated

RESPONSE:  Corrected. It was a formatting issue. Thank you

-Section 3 and 4 should be merged with section 2.Materials and Methods.

RESPONSE:   Thank you for this, we have merged Section 2 with Section 3, as suggested

- Section 4.1. is well known, perhaps it could be schematised.

RESPONSE:  please consider we would prefer not to reduce further this section, to give a little more explanation to the reader about the method used in FEM

- Line 336 and 338, cannot be Table 1, should be Table 5.

RESPONSE:  Corrected. It was a formatting issue. Thank you

  • Line 379, Correct Figure 101) and the end of the sentence.

RESPONSE:  Corrected. It was a formatting issue. Thank you

There is no final discussion of the numerical results and at no point are the results compared with other research. This section should be corrected.

RESPONSE:  This section has been fully reviewed and corrected “Comparing the experimental results with the numerical ones is a challenging and problematic task: clearly, numerical results are governed by the mechanical parameters given in Tables 4 and 5. These values have been found in the scientific literature, with no direct relation with the mechanical properties of the brickwork masonry used in this experimental work. This is a significant limitation of this investigation. However, the aim of the numerical analysis was to verify if the use of a simplified numerical approach, only based on mechanical parameters found in the literature, was able to provide ac-acceptable results useful in structural design and give the professional world an indication of the error this very simplified numerical method can produce. “

Figures 15 and 16 shows the comparison between numerical and experimental results in terms of shear behavior. It can be noted that a large shear strength error was noted for unreinforced and PP-reinforced walls (about 30%), while for FC-reinforced walls the error was significantly smaller (about 15%).

  • Line 490. Figure 136

RESPONSE:  Corrected. It was a formatting issue. Thank you

The limitations found in this research should be included at the end of the conclusions.

RESPONSE: thank you for this question. The following was added “Reader should be alerted about the drawbacks of the proposed retrofitting methods. The main limitation on the fullest possible use of small-diameter steel wires in ferrocement retrofit is the risk of corrosion. However, this could be mitigated by using high-strength stainless steel grids. For CFRP and PP applications, fiber and matrix degradation in exposed environments (UV radiation, extreme and harmful weather events, rain and humidity) could be significant. More tests will be necessary to assess the long-term effectiveness of the proposed retrofitting methods.”

Round 2

Reviewer 1 Report

1.In Figure 2a, it can be seen that the threaded bolts are not evenly arranged, does it affect the test results?

The reply is ok.

2. Part of the picture number is wrong, as shown in Figure 90, 101, 125 and 136;

As shown in the paper, the picture number is not corrected.

3.From Section 4 of the article, it can be seen that the test has been simulated in advance, but why there is no simulation effect diagram, especially the corresponding cracking simulation diagram. Is the conclusion given too simple? The simulation effect is not ideal, so what is the meaning of this part?

We all know that the experiment is discrete, and the simulation cannot be predicted more accurately, but it can know the trend of the test. With the simulation effect diagram, the article data in 4.1 can be verified. So I hope the author try to supplement part model diagrams.

4.In Figure 14, the URM analysis is "the average shear strength of the URM panels was 0.153 MPa, with a maximum of 0.177 MPa reached for W1 and a minimum of 0.129 MPa for W2.", but the maximum shear stress of the URM specimen in Figure 14 shows W3, and the maximum test load in Table 6 is also W3, which is inconsistent with the description, and needs to be verified.

As shown in the figure 14 and table 6, the description is not corrected, the average shear strength of the URM panels was 0.133 MPa, with a maximum of 0.153 MPa reached for W3 and a minimum of 0.117 MPa for W1.".

5.It can be seen from Figure 14 that the results of the same group of specimens show a certain degree of discreteness, but the results of different groups of specimens have certain regularity. Among the three specimens in the same group, the results of two specimens show consistency, but there is always one showing a large deviation from the others. Are there other factors in the test?

The reply is ok.

6. In Figure 14, W-8 value reaches its peak and remains stable. Is it wrong?

The reply is ok.

7.The analysis of the reinforcement mechanism is still lacking, and the feasibility of reinforcement is only given from the comparison of test phenomena and data levels, and the depth of content is insufficient.

The reply is not satisfactory.

The English writing level of the article is relatively high, which can well display the viewpoints of the paper, and the layout is relatively reasonable.

Author Response

Please note the attached file with a point-by-point response.

Reviewer 2 Report

Accept.

Author Response

Firstly, the authors would like to thank the reviewers for the valuable comments and suggestions. We believe that thanks to these suggestions the quality of the manuscript is considerably improved.

Reviewer 3 Report

The authors have made all the proposed changes and have significantly improved the submitted manuscript.

Author Response

(The authors gave the same response as above.)
